

# Estimation of simultaneous equation models by backpropagation method using stochastic gradient descent

Belén Pérez-Sánchez[1], Carmen Perea[1], Guillem Duran Ballester[2] and Jose J. López-Espín[1]

[1] Center of Operations Research, Universidad Miguel Hernández de Elche, Elche, Alicante, Spain
[2] Fragile Tech, Palma de Mallorca, Mallorca, Spain

## ABSTRACT

Simultaneous equation model (SEM) is an econometric technique traditionally used in economics but with many applications in other sciences. This model allows the bidirectional relationship between variables and a simultaneous relationship between the equation set. There are many estimators used for solving an SEM. Two-steps least squares (2SLS), three-steps least squares (3SLS), indirect least squares (ILS), *etc.* are some of the most used of them. These estimators let us obtain a value of the coefficient of an SEM showing the relationship between the variables. There are different works to study and compare the estimators of an SEM comparing the error in the prediction of the data, the computational cost, *etc.* Some of these works study the estimators from different paradigms such as classical statistics, Bayesian statistics, non-linear regression models, *etc.* This work proposes to assume an SEM as a particular case of an artificial neural networks (ANN), considering the neurons of the ANN as the variables of the SEM and the weight of the connections of the neurons the coefficients of the SEM. Thus, backpropagation method using stochastic gradient descent (SGD) is proposed and studied as a new method to obtain the coefficient of an SEM.

## INTRODUCTION

Simultaneous equation models (SEM) are used as a statistical technique encompassing a multitude of equations, which are solved concurrently to examine the intricate relationships between multiple variables. These models find widespread use in various fields, such as economics, finance, engineering, and social sciences. For example, an SEM is used in tax research employing provincial-level data spanning the period of 2001–2014 in Indonesia to investigate the effects of fiscal decentralization on regional income inequality (*Siburian, 2019*). Similarly, the study and exploration of an SEM connecting employment and mental health has been conducted (*Steele, French & Bartley, 2013*). The peer effects in casino gambling behavior (*Park & Manchanda, 2015*), the interaction between individuals' health risk perception and betel chewing habits in Taiwan (*Chen et al., 2013*), the effects of repetitive iodine thyroid blocking on the development of the fetal brain and thyroid

Corresponding author
Belén Pérez-Sánchez,
m.perezs@umh.es

in rats (*Cohen et al., 2019*), the impact of foreign trade on energy efficiency within China's textile industry (*Zhao & Lin, 2019*), as well as the assessment of biomass energy consumption, ecological footprint through FDI, and technological innovation in B&R economies (*Yasmeen et al., 2022*) are examples of applications of SEMs. The estimation of an SEM entails identifying the variable values that simultaneously satisfy all equations within the model. Full information maximum likelihood (FIML) or three-stage least squares (3SLS), can be used as system methods, and compared with more widely used methods like ordinary least squares (OLS), indirect least squares (ILS), and the two-stage least squares (2SLS) (*Gujarati & Porter, 2004*). All these methods are set in classical statistics but other paradigms can be considered as the Bayesian Statistics. Some estimation techniques that appear with this new econometric approach of SEM are the Bayesian method of moments (BMOM) and the minimum expected loss (MELO) estimator (*Zellner, 1997*) or the methods used by *Chao & Phillips (2002)*, and *Kleibergen & Dijk (1998)*. This attempt to more realistic models approach leads to analytical difficulties as Steel's work documents (*Steele, French & Bartley, 2013*), and, as a result, new algorithms have appeared to facilitate the calculation of the posterior distributions in Bayesian models. One of the most important problems here is that the Bayesian analysis of SEM introduces an inevitable degree of complexity due to the prior specification of the distributions as well as to the posterior obtaining of the distributions.

Artificial neural network (ANN) is a machine learning model inspired by the human brain's structure and function (*Goodfellow, Bengio & Courville, 2016*). ANNs consist of layers of interconnected nodes that process information and learn to make predictions or decisions. They are used in a wide range of applications, including image encryption (*Mohanrasu et al., 2023*), natural language processing (*Khurana et al., 2023*), hydrogen production (*Abdelkareem et al., 2022*), medical diagnosis and analysis (*Surianarayanan et al., 2023*), autonomous vehicles and robotics (*Ali et al., 2023*), cybersecurity for intrusion detection and malware detection (*Bharadiya, 2023*).

The applications of ANN to SEM are diverse, as illustrated by several key studies. *Kumar (1991)* presents an approach to formulating and estimating an SEM of the US economy as a neural network problem. He concludes that this new approach is promising, albeit with reservations due to the small size of the problem. A study shows the comparison results when estimating the Klein I model through statistical methods against a neural network with two hidden layers, in which the feasibility and good results of applying ANN to this type of problem are highlighted (*Brennan & Marsh, 1992*). A similar work by *Caporaletti et al. (1994)*, also solves the Klein economic problem posed as an SEM of two equations by comparing the results of estimating by classical methods such as 2SLS and 3SLS with an ANN, obtaining very similar results between the different methods. On the other hand, *Ma et al. (2021)* studies the diffusion of scientific articles from the academic world through social networks using SEM and ANN. An infographic of publications on applications of ANN to SEM, ANN and SEM is shown in Fig. 1.

To adjust the connection weights of the ANN, the backpropagation method (BM) is used compensating for each error found during learning. BM obtains the derivative (the gradient) of the cost function associated (error function) in each iteration with respect to

**Figure 1 Timeline of publications on applications of ANN to SEM, SEM and ANN.**

the weights. The weight updates can be obtained using gradient descent or other methods, such as extreme learning machines, training without backtracking, weightless networks, *etc.* (*Ollivier, Tallec & Charpiat, 2015*).

The main contribution of this paper is to consider an SEM as a particular case of an ANN and thus, by using the techniques to solve the ANN, *i.e.,* obtaining the weight of the edges from the neurons, obtaining the coefficients of the parameters in the SEM. Therefore, the paper proposes a new algorithm to estimate the SEM by considering an SEM as an AN.

The document is structured as follows. 'Simultaneous Equation Mode' presents the model and discusses some methods for estimating linear SEMs. 'Simultaneous Equation Model as a Particular Case of Artificial Neural Network' explains how an SEM is represented as a particular case of an ANN. 'Stochastic Gradient Descent' provides a review of stochastic gradient descent. The experimental design and findings of the comparative study are presented in the 'Experimental Study'. Finally, the 'Conclusions' offers concluding comments.

## SIMULTANEOUS EQUATION MODEL

Considering $m$ interdependent or endogenous variables which depend on $k$ independent or exogenous variables, and supposing that each endogenous variable can be expressed as a linear combination of some of the rest endogenous variables, and the exogenous ones, adding a white noise variable that represents stochastic interference. Thus, a linear SEM (*Gujarati & Porter, 2004*) is:

$$
\begin{aligned}
Y_1 = & \ B_{1,2}Y_2 + B_{1,3}Y_3 + \cdots + B_{1,m}Y_m + \Gamma_{1,1}X_1 + \cdots + \Gamma_{1,k}X_k + u_1 \\
Y_2 = & \ B_{2,1}Y_1 + B_{2,3}Y_3 + \cdots + B_{2,m}Y_m + \Gamma_{2,1}X_1 + \cdots + \Gamma_{2,k}X_k + u_2 \\
& \ \vdots \\
Y_m = & \ B_{m,1}Y_1 + B_{m,2}Y_2 + \cdots + B_{m,m-1}Y_{m-1} + \Gamma_{m,1}X_1 + \cdots + \Gamma_{m,k}X_k + u_m
\end{aligned}
\tag{1}
$$

where $B \in \mathbb{R}^{m \times m}$ and $\Gamma \in \mathbb{R}^{m \times k}$ are matrices of coefficients, and $x$, $y$ and $u$ are exogenous, endogenous and white noise variables, which are vectors of dimension $n$, being $n$ the sample size. Some coefficients of $B_{i,j}$ and $\Gamma_{k,r}$ are zero, and are known *a priori*. The equation can

be represented in matrix form as:

$$YB^T + X\Gamma^T + U = 0 \qquad (2)$$

where $Y = (Y_1, \dots, Y_m)$, $X = (X_1, \dots, X_k)$ and $U = (U_1, \dots, U_m)$.

SEM can be representing by the reduce form as

$$Y = X\Pi + V \qquad (3)$$

where $\Pi = -\Gamma^T(B^T)^{-1}$, and $V = -U(B^T)^{-1}$.

Solving the model is equivalent to obtaining an estimation of $B$ and $\Gamma$ in Eq. (2) from a representative sample of the data variables $X$ and $Y$. Based on classic inference, the estimation methods can be set by limited and full information methods. Limited information methods estimate each of the equations without using the information contained in the rest of the model, *i.e.,* only considering both the endogenous and exogenous variables included in the equation. Ordinary least squares (OLS), indirect least squares (ILS), and two-stage least squares (2SLS) are examples of these methods (*Gujarati & Porter, 2004*).

Full information methods consider joint estimation of the whole model in the structural form. These methods require the specification of all the equations, and all of them must be specified to have a solution. In general, they are more asymptotically efficient than the others since they incorporate all the information of the system. However, the drawback is that inconsistent estimates may be generated if any equation is incorrectly specified. Examples of these kinds of methods are FIML or 3SLS (*Gujarati & Porter, 2004*).

On the other hand, Bayesian inference does not use sampling assumptions. However, it introduces a high degree of complexity due to the prior specification of the distribution and the obtaining of the posterior distribution. Some techniques are the BMOM developed by *Zellner (1998)*, or the methods used by *Chao & Phillips (2002)*, who study the behavior of posterior distributions under the Jeffreys prior in a simultaneous equations model (*Chao & Phillips, 2002*). Geweke developed general methods for Bayesian inference with non-informative reference prior in the model, based on a Markov chain sampling algorithm, and procedures for obtaining predictive odds ratios for regression models with different ranks *Geweke (1996)* and *Kleibergen & Dijk (1998)* that solve Bayesian SEM using reduced rank structures (*Kleibergen & Dijk, 1998*).

The Markov Chain Monte Carlo development has been key in making the computation of large models that require integration over hundreds or even thousands of unknown parameters possible. The Metropolis–Hastings algorithm and the Gibbs sampling (*Gelman et al., 2015*) are examples. A recent study was carried out to optimize the parameters $K_1$ and $K_2$ from BMOM in order to minimize the Akaike Information Criteria (AIC). Furthermore, this method, called $Bmom_{OPT}$, obtains estimated parameters with minimal error (*Pérez-Sánchez et al., 2021*).

## SIMULTANEOUS EQUATION MODEL AS A PARTICULAR CASE OF ARTIFICIAL NEURAL NETWORK

The goal of this work is to study stochastic gradient descent as a method of obtaining the coefficient of an SEM assuming that this SEM is a particular case of an ANN, considering the neurons of the ANN as the variables of the SEM and the weight of the connections of the neurons the coefficients of the SEM.

For example, considering a model with two endogenous variables, $y_1$ and $y_2$, and four exogenous variables, $x_1$, $x_2$, $x_3$ and $x_4$, with the next structure as an example:

$$y_1 = B_{1,2}y_2 + \Gamma_{1,1}x_1 + \Gamma_{1,2}x_2 + \Gamma_{1,4}x_4 + u_1$$
$$y_2 = B_{2,1}y_1 + \Gamma_{2,1}x_1 + \Gamma_{2,2}x_2 + \Gamma_{2,3}x_3 + \Gamma_{2,4}x_4 + u_2$$

(4)

This SEM 4 can be interpreted as a multi-layer perceptron (MLP) (*Kruse et al., 2022*), a type of artificial neural network composed of an input layer, one or more hidden layers, and an output layer. Figure 2 shows a simple one-layer ANN structure. In this analogy, the endogenous variables $y_1$ and $y_2$ are like neurons in the output layer, while the exogenous variables $x_1$, $x_2$, $x_3$, and $x_4$ are like neurons in the input layer. The relationships between the endogenous and exogenous variables in the SEM, represented by the coefficients, are similar to the synaptic weights in an MLP.

The relationship between the endogenous variables in the SEM reflects the interconnected nature of neurons in an MLP, where $y_1$ influences $y_2$ and vice versa. The error terms $u_1$ and $u_2$ can be considered as noise in signal transmission, similar to the imperfections and fluctuations encountered in the input and output data of an ANN.

In summary, the structure and interactions in an SEM with endogenous and exogenous variables are similar to those in an MLP. Both models capture complex relationships between variables and can be used to analyze patterns and make predictions.

However, ANNs need large amounts of data for training, which can be a problem for SEM applications with limited data. Additionally, the complexity of ANNs, especially with hyperparameter tuning and model design, can present challenges not found in traditional methods.

## STOCHASTIC GRADIENT DESCENT

Gradient descent is a classic and widely used technique in the field of machine learning and artificial neural networks for minimizing cost functions. Its application has extended beyond the realm of machine learning and has been adapted to solve various optimization problems in the fields of electronics (*Nawaz et al., 2019*), telecommunications (*He et al., 2022*), fluid dynamics (*Chen et al., 2022*), *etc.*

In essence, gradient descent is an iterative algorithm that seeks to find the local (or global) minimum of a function by making gradual adjustments in the direction opposite to the gradient of the function at the current point. The gradient represents the direction of the steepest ascent of the function, so moving in the opposite direction of the gradient results in reducing the value of the function with each iteration. Over iterations, the algorithm converges toward a minimum, which translates to an optimal or approximate solution to

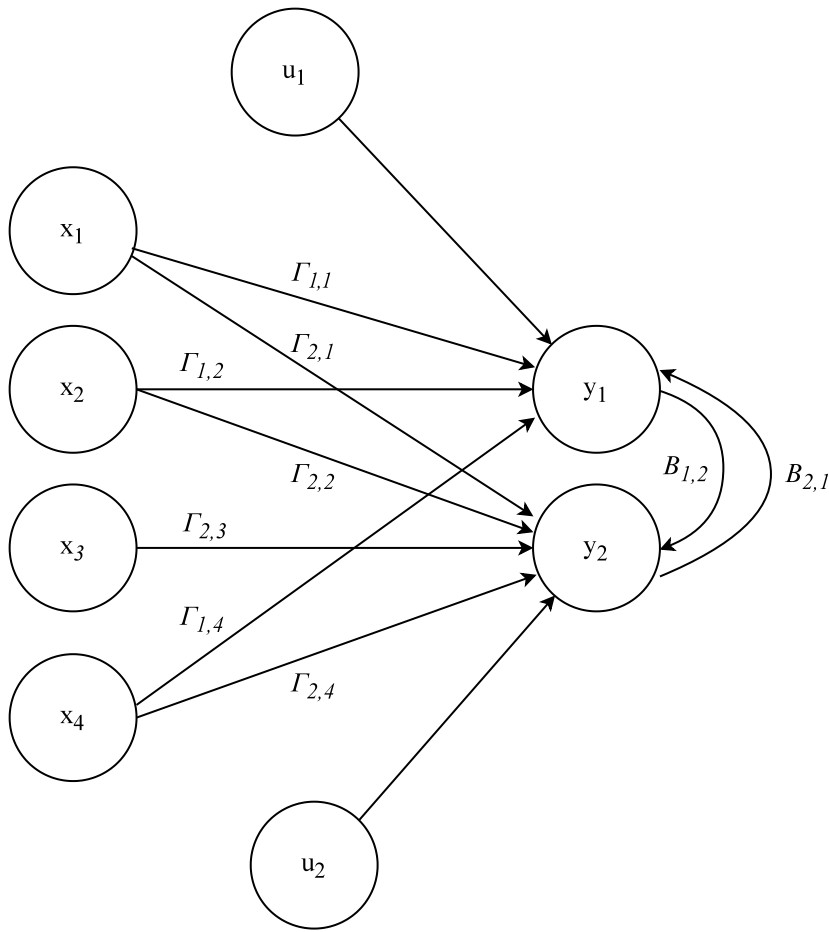

**Figure 2** Representation of SEM as an ANN.

the problem at hand. More details about the gradient descent technique can be found in *Nocedal & Wright (2006)*.

Moreover, despite being a classic technique in the field of optimization, gradient descent maintains its relevance and vitality today. Its durability is largely due to the constant attention it receives from researchers and scientists across various fields. In the era of artificial intelligence and machine learning, where optimization is essential, gradient descent remains a cornerstone. Numerous recent studies have extensively explored its strengths and weaknesses (*Ahn, Zhang & Sra, 2022*), adapting it to a wide range of applications (*Zhang, Qiu & Gao, 2023*), as well as enriching its theoretical understanding (*Jentzen & Von Wurstemberger, 2020*).

Furthermore, over time, it has given rise to several variations and adaptations that are tailored to different contexts and types of problems. Stochastic gradient descent (SGD), gradient descent with momentum, adaptive gradient descent, mini-batch gradient descent, and adaptive learning rate are among the most noteworthy variants (*Ruder, 2016*).

The stochastic gradient descent method, instead of computing the gradient over the entire dataset, calculates the gradient using only a random sample in each iteration. This makes the algorithm faster and more scalable, which is beneficial when working with massive datasets. Some parameters of this algorithm are briefly introduced: First, the initial seed is used to initialize random number generators involved in random sample selection from the dataset or in model parameter initialization. This is important to ensure experiment reproducibility, as using the same initial seed will yield the same results in subsequent runs.

Second, the learning rate is a critical parameter, as it controls the step size at which the algorithm updates the parameters of the model in each iteration. A higher learning rate can expedite convergence but may also introduce instability. Conversely, a lower learning rate can obtain convergence but might require more iterations. Finally, the batch size is the number of training examples used in each iteration of SGD. A small batch introduces more variability in updates, which can help escape local minima but may increase noise in gradient estimates. On the other hand, a large batch can provide more stable gradient estimates but may require more computational resources.

In this work, the SGD method was chosen due to its highly versatile and efficient variant of the classic gradient descent that is used in a variety of optimization problems, especially in the field of machine learning and ANN. It can be considered that simultaneous equation models can, in some way, be regarded as a particular case of ANN. Among the advantages of SGD, it can be highlighted that it is scalable to handle large problems, efficient in the use of small batches of data instead of the complete dataset, and it improves convergence and stability by allowing for the adjustment of the learning rate.

## EXPERIMENTAL STUDY

Multiple SEMs have been generated varying the number of endogenous and exogenous variables and the sample size and have been solved using both statistical inference and SGD method. In this comparative study, the 2SLS method has been employed given its widespread utilization in resolving SEM due to its simplicity and its goods outcomes.

The SEMs Eq. (2) have been generated as follows: Matrices B and $\Gamma$ have been randomly obtained from a Uniform distribution in [0, 10] and X is a matrix obtained from a multivariate normal distribution. Matrix Y has been calculated from Eq. (3), where $V$ follows a normal distribution with parameters $\mu = 0$ and $\sigma = \{0.1, 1.0\}$. For each type of SEM, a datasets of 100 and 1,000 observations has been created, varying across two levels of variability defined by $\sigma$ values.

As a metric for assessing the accuracy of predictive models, the mean squared error (MSE) has been used. The mathematical expression of MSE for the SEM can be represented as: $\frac{1}{n \times m} \sum_{j=1}^{m} \sum_{i=1}^{n} (Y_{ij} - \hat{Y}_{ij})^2$, where $n$ denotes the size of the test or validation sample, $m$ represents the number of endogenous variables of the SEM, and $Y$ and $\hat{Y}$ symbolize the endogenous variable and its corresponding predictive value, respectively.

Table 1 presents the average and standard deviation (std) of the MSE obtained from 10 repetitions of the experiment using a dataset of size 1,000. The table has been divided

**Table 1  Average and standard deviation of MSE obtained by 2SLS and SGD.**

| SEM | $\sigma$ | Method | | Average | Std |
|---|---|---|---|---|---|
| m=2<br>k=4 | 0.1 | 2SLS | | 0.225469 | |
| | | SGD | 2sls | 0.066712 | 0.000018 |
| | | | rnd | 0.066712 | 0.000018 |
| | 1.0 | 2SLS | | 0.007806 | |
| | | SGD | 2sls | 0.007819 | 0.000010 |
| | | | rnd | 0.007819 | 0.000010 |
| m=10<br>k=20 | 0.1 | 2SLS | | 1.442922 | |
| | | SGD | 2sls | 0.204201 | 0.000138 |
| | | | rnd | 0.204709 | 0.000068 |
| | 1.0 | 2SLS | | 0.815662 | |
| | | SGD | 2sls | 0.233558 | 0.000070 |
| | | | rnd | 0.233765 | 0.000063 |
| m=20<br>k=40 | 0.1 | 2SLS | | 3.046122 | |
| | | SGD | 2sls | 0.277536 | 0.000302 |
| | | | rnd | 0.246335 | 0.000190 |
| | 1.0 | 2SLS | | 3.143916 | |
| | | SGD | 2sls | 0.336215 | 0.000089 |
| | | | rnd | 0.336976 | 0.000122 |

into three sections, one for each type of SEM defined in terms of m and k, which represent the number of endogenous and exogenous variables, respectively. Within each section, the table is further divided into two subsections according to the variability ($\sigma$) and the method used for solving the SEM (2SLS or SGD). For the SGD case, the row has been subdivided, one for each initial seed.

The 2SLS method has been used on the entire dataset to estimate the model parameters and, the MSE has been calculated using random samples of size 300. Only averages are provided, since the standard deviation was almost zero.

The SGD method has been obtained by using two initial seeds for the gradient descent process: one is the solution obtained by 2SLS method ('2sls'), and the other is a generated randomly ('rnd'). Seventy percent of the data has been used for training, while the remaining 30 percent has been allocated for validation. Furthermore, two learning rates, 0.01 and 0.00001, have been used in the experiment. However, since insignificant discrepancies were observed between them, only the results obtained using 0.01 are presented in the table. The model training involved a batch size of 32, which represented randomly chosen training examples used in each weight update to compute the gradient. The process consisted of 5000 iterations when the learning rate was set to 0.01, and 10,000 iterations were performed with a learning rate of 0.00001. Additionally, gradient values were restricted to 0.5 during each step (gradient clipping by value), although it showed negligible effects on the training process.

Upon reviewing the Table 1 results, a comparison is made between the minimum MSE values obtained using the 2SLS and SGD methods across the different models for a variability of 0.1. In the $m = 2\ k = 4$ model, the 2SLS method produced an MSE of

0.225469, while SGD produced 0.066712 (seed '2sls' and 'rnd'). In the $m = 10$ k $= 20$ model, 2SLS obtained an MSE of 1.442922, while SGD resulted in 0.204201 (seed '2sls') and 0.204709 (seed 'rnd'). Finally, in the $m = 20$ k $= 40$ model, 2SLS recorded an MSE of 3.046122, while SGD showed 0.277536 (seed '2sls') and 0.246335 (seed 'rnd'). These results suggest a consistent trend across different models with lower MSE values obtained using the SGD method compared to the 2SLS method. Comparing the results obtained through the SGD method, the MSE values coincide for both seeds in the case of the small model. In the intermediate model, the minimum MSE value is found with the '2sls' seed, although the one obtained with 'rnd' closely follows. In the largest model, the minimum is achieved with the 'rnd' seed. This shows that using the '2sls' seed can improve the prediction of endogenous variables.

For the dataset with higher variability ($\sigma = 1.0$), in the $m = 2$ k $= 4$ model, an MSE of 0.007806 is obtained with 2SLS and 0.007819 with SGD (seed '2sls' and 'rnd'). In the $m = 10$ k $= 20$ model, 2SLS obtains 0.815662, while SGD results in 0.233558 (seed '2sls') and 0.233765 (seed 'rnd'). Finally, in the $m = 20$ k $= 40$ model, 2SLS obtain a value of 3.143916, whereas SGD results in 0.336215 (seed '2sls') and 0.336976 (seed 'rnd'). Comparing the results within the SGD method, the MSE values obtained using the '2sls' seed are lower than those obtained using 'rnd', but they are very close to each other. Therefore, except for the $m = 2$ k $= 4$ model with a variability of 1.0, the minimum MSE values were obtained using the SGD method across the remaining studied cases. When comparing the initial seeds, in all cases, the minimum MSE value has been obtained using '2sls' as the initial seed in the SGD method.

A dataset of size 100 has also been used in the experiment, with 70 for SGD training and 30 for validation. The 2SLS method utilized the entire dataset for coefficient estimation and 30 for calculating the MSE. The results are then compared with those obtained from a dataset of size 1,000 and presented in bar charts (Figs. 3 and 4). Both figures display the MSE using each method for the two largest SEMs and based on the two $\sigma$ values. The analysis of these charts shows that when employing a small dataset of size 100, there are no significant variations in the MSE results compared to a larger dataset of size 1,000. This observation suggests that the trained models do not exhibit overfitting, as their performance on the validation dataset shows no notable decreases. This robust behavior underscores the ability of the models to efficiently capture meaningful patterns in the data without depending on the training dataset.

## CONCLUSIONS

Simultaneous equation models are employed in situations where a bidirectional relationship exists between variables, a common scenario in various research fields.

One of the objectives of this work is to enhance prediction accuracy by applying a novel technique to derive the parameters of an SEM as a particular case of an ANN. Specifically, the stochastic gradient descent technique is utilized while preserving the original architecture of the SEM. A comparative analysis was conducted to estimate the coefficients of three types of SEM, utilizing the 2SLS method and SGD method with variations in parameters such as training data, learning rate, and the initial seed. The results indicate that the SGD

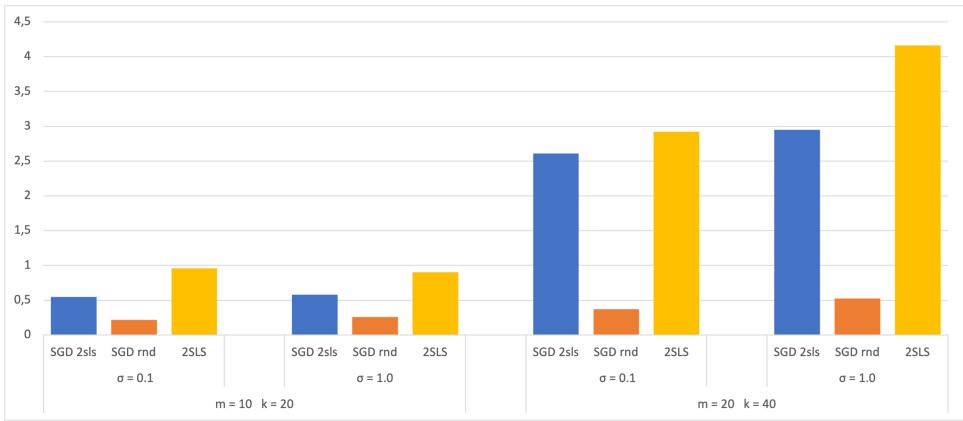

**Figure 3  Average of MSE for training data 70.**

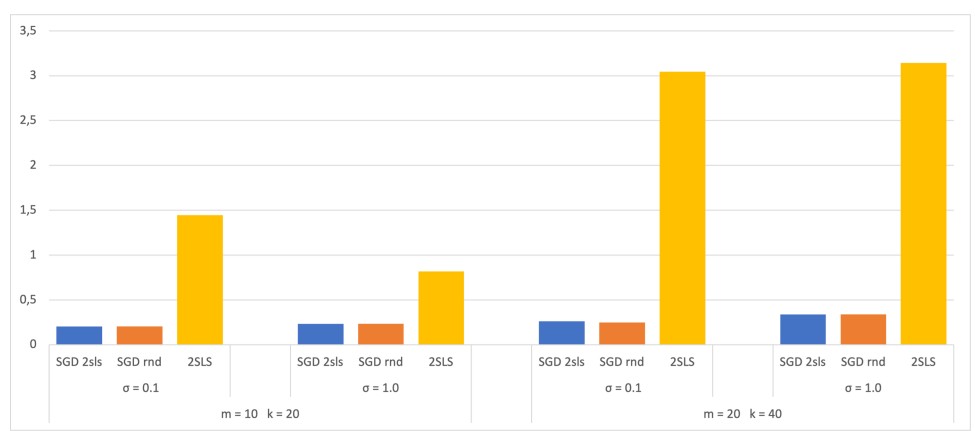

**Figure 4  Average of MSE for training data 700.**

method achieves better results in terms of minimising prediction error when using the 2SLS solution as the initial seed, with almost no differences found between different learning rates or the training data sizes. Although computation time has not been studied yet, the stochastic version of the method is shown to expedite problem resolution.

Aware of the limitations of our work, future research will involve expanding the dataset and performing additional experiments to generalize the results. Datasets with different sizes and variability will be considered, as well as analyzing the impact of different hyperparameter configurations on SGD performance to obtain more accurate estimates. Additionally, real-world data from health sciences and other fields will be used.

### Funding

The authors received no funding for this work.

## Competing Interests

Guillem Duran Ballester is employed by Fragile Tech. The authors declare there are no competing interests.

## Author Contributions

- Belén Pérez-Sánchez conceived and designed the experiments, performed the experiments, analyzed the data, prepared figures and/or tables, and approved the final draft.
- Carmen Perea performed the experiments, analyzed the data, authored or reviewed drafts of the article, and approved the final draft.
- Guillem Duran Ballester performed the experiments, performed the computation work, prepared figures and/or tables, and approved the final draft.
- Jose J. López-Espín conceived and designed the experiments, analyzed the data, authored or reviewed drafts of the article, and approved the final draft.

## Data Availability

The code is available at GitHub and Zenodo:

- https://github.com/Guillemdb/simultaneous-equations

- Guillem DB. (2024). Guillemdb/simultaneous-equations: Create DOI (v0.0.2). Zenodo. https://doi.org/10.5281/zenodo.13303985

The raw data is available in the Supplemental File.

## Supplemental Information

Supplemental information for this article can be found online at http://dx.doi.org/10.7717/peerj-cs.2352#supplemental-information.

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
