# Peer review of "Estimation of simultaneous equation models by backpropagation method using stochastic gradient descent"

_PeerJ Computer Science, doi:10.7717/peerj-cs.2352_

## Round 0.1 · original submission · Major Revisions

· Academic Editor

Major Revisions

Please, consider the reviewers' comments to improve the quality of the submission.

Reviewer 1 ·

Basic reporting

All comments have been added in detail to the last section.

Experimental design

All comments have been added in detail to the last section.

Validity of the findings

All comments have been added in detail to the last section.

Additional comments

Review Report for PeerJ Computer Science
(Estimation of simultaneous equation models by backpropagation method using stochastic gradient descent)

1. Within the scope of the study, the stochastic gradient descent optimization technique used in the backpropagation part of artificial neural networks was analyzed and proposed to obtain the coefficient of a simultaneous equation model.

2. In the introduction section, what simultaneous equation models are and where they are used, some estimation techniques, artificial neural networks and simultaneous equation model applications are basically mentioned. In this section, in order to emphasize the importance of the study and its place in the literature, it is suggested to add a literature table regarding both artificial neural networks and simultaneous equation model applications.

3. In this study where artificial neural networks are associated with the simultaneous equation model, the difference of the study from the literature, its originality point and main contributions to the literature should be stated in more detail and clearly in bullet points at the end of the introduction section.

4. In the second part of the study, a linear simultaneous equation model structure with its equations and variables, and the developments up to the present day are sufficiently mentioned.

5. In the third part of the study, the representation of the simultaneous equation model in an artificial neural network is explained. Although the explanations and examples in this section are basically sufficient, it is recommended to make detailed explanations, especially for multilayer artificial neural networks.

6. When the information and experimental study regarding stochastic gradient descent are examined, the obtained dataset, metrics and results are generally appropriate. However, it is recommended to observe the analysis of the results for higher amounts of data and to expand the results by increasing the amount of dataset and conducting additional experiments in order for the study to stand out more.

7. In the conclusion section, more detailed information can be provided for future works, and different application suggestions can be made regarding the application of the simultaneous equation model in artificial neural networks.

As a result, although the study is at a certain level in terms of the subject and application originality, it is recommended to pay attention to the sections mentioned above.

Reviewer 2 ·

Basic reporting

The introduction provides good context on SEMs and estimation methods, but could benefit from a clearer statement of the paper's specific contributions and novelty.

The explanation of how SEMs map to ANNs in Section 3 is clear and helpful. Consider adding a brief discussion of potential limitations of this analogy.

There are some grammatical errors and unclear phrasings throughout that should be carefully edited (e.g. "an experimental study is conducted to estimate the coefficients of SEMs, applying the 2SLS method and SGD method with variations in parameters such as training data, learning rate, and the initial seed.")

Experimental design

The experimental design in Section 5 is thorough, but adding at least one real-world dataset to complement the simulations would strengthen the paper.

Consider comparing to at least one other modern SEM estimation technique beyond just 2SLS to better contextualize the performance of SGD.

Validity of the findings

The results show promise for the SGD approach, especially on larger models. However, more analysis of why SGD outperforms 2SLS would strengthen the paper. Are there specific characteristics of SEMs that make SGD particularly suitable?

The conclusion could be strengthened by discussing limitations of the approach and potential directions for future work.

---

## Round 0.2 · accepted · Accept

· Academic Editor

Accept

Thank you for the hard work and effort in addressing the reviewers' comments to improve the article; it is reflected in the final result.

Reviewer 1 ·

Basic reporting

All comments have been added in detail to the last section.

Experimental design

All comments have been added in detail to the last section.

Validity of the findings

All comments have been added in detail to the last section.

Additional comments

Thanks for the revision. The answers given are sufficient.